# An Integrated Safety, Health and Environmental Management Capability Maturity Model for Construction Organisations: A Case Study in Ghana

Millicent Asah-Kissiedu [1], Patrick Manu [2], Colin Anthony Booth [3,*], Abdul-Majeed Mahamadu [3]
and Kofi Agyekum [4]

1   Department of Environmental Management and Technology, Koforidua Technical University,
    Koforidua KF981, Ghana; mokumih@ktu.edu.gh
2   Department of Mechanical, Aerospace and Civil Engineering, The University of Manchester,
    Manchester M13 9PL, UK; patrick.manu@manchester.ac.uk
3   Department of Architecture and the Built Environment, University of the West of England,
    Bristol BS16 1QY, UK; Abdul.Mahamadu@uwe.ac.uk
4   Department of Construction Technology and Management, Faculty of Built Environment, College of Art and
    Built Environment, Kwame Nkrumah University of Science and Technology, Kumasi AK384, Ghana;
    kagyekum.cap@knust.edu.gh
*   Correspondence: Colin.Booth@uwe.ac.uk; Tel.: +44-117-328-3998

**Abstract:** Safety, health and environmental (SHE) management is becoming a priority as construction companies (i.e., contractors) strive to reduce construction accidents and negative environmental impacts, conform to regulatory requirements, and sustain their competitiveness. Consequently, construction firms are expected to adopt and implement innovative SHE management systems to mitigate SHE risks effectively and efficiently. For construction firms to effectively do this, they need to have the adequate capability in respect of integrated SHE management. However, there is limited empirical insight regarding the integrated SHE management capabilities of construction companies. Furthermore, there is limited insight regarding the mechanisms for ascertaining the integrated SHE management capability of construction companies to guide such organisations towards SHE management excellence in their operations. Drawing on the capability maturity model integration (CMMI) concept, this study, by applying expert reviews (i.e., Delphi technique and the design methodology for capability maturity grids), developed an integrated Safety, Health and Environmental Management Maturity Model (iSHEM-CMM). The model offers capability maturity assessment on a five-level scale within five thematic categories and 20 integrated SHE management capability attributes. Based on an industrial validation by construction professionals, it is concluded that the maturity model is a useful assessment framework or tool for industry stakeholders, particularly construction firms, to evaluate the status of their current SHE management capabilities, identify strengths and improvement areas, and accordingly prioritise strategies/actions for improving their SHE management. Furthermore, clients who appoint construction companies could use the model as part of prequalification arrangements in selecting construction companies with an adequate SHE management capability.

**Keywords:** capability attribute; capability maturity model; construction; integrated safety; health and environmental management

## 1. Introduction

The construction sector remains one of the key generators of adverse environmental impacts and is among one of the highest contributors of work-related accidents, resulting in injuries, fatalities, and illnesses [1–3]. For instance, in the USA, the construction sector accounted for over 800 worker-related deaths in 2019 [4]. Moreover, the UK construction sector recorded the highest number of fatal injuries in 2020/21 [5]. The construction sector

in India, having only 7.5% of the total world labour force, also contributes 16.4% of fatal occupational accidents worldwide [6].

Furthermore, the sector accounts for a significant consumption of natural resources and energy. Estimates indicate that buildings and construction together account for 36% of final energy use, 16% of natural water, 39% of $CO_2$ emissions, 40% of the waste produced and 50% of all raw materials extracted [7,8]. With the volume of construction output projected to grow by more than 85% globally by 2030 [9], the impact of construction operations on the environment and workers' safety and health would be far-reaching if nothing were to be done about it. The socio-economic impacts arising from these negative environmental impacts, injuries, illnesses, and fatalities [3,10] have triggered several efforts to address the poor status of SHE management in construction. One of the prominent initiatives to address the SHE situation in construction is the implementation of management systems, particularly environmental management systems (EMSs) and safety and health management systems (SHMSs) in construction to manage SHE risks with maximum effectiveness and minimum bureaucracy [11]. This could be beneficial in reducing the number of fatalities, injuries, illnesses and potentially negative environmental impacts, leading to better SHE performance outcomes within the construction sector.

Like other countries [4–6], in Ghana, the construction industry accounts for a high number of occupational accidents and deaths as well as work-related illnesses [12]. The construction industry in Ghana is also noted for its constant degradation, pollution, substantial raw materials and energy consumption, which negatively impact the development of the country [12]. Despite these negative impacts, Agyekum et al. [13] reported that the high-risk nature of the industry, the weak institutional structures for implementing SHE standards, and laxity in the enforcement of safety and environmental legislations on construction sites have impeded the implementation of SHE standards. Due to these lapses, there has been a need to implement proactive and systematic methods that have the potential to prevent accidents and negative environmental impacts on construction sites, and that will further assist construction companies to effectively improve SHE performance outcomes in the industry. Unfortunately, the uptake of a prominent approach like the implementation of SHE management systems in the Ghanaian construction industry is low [12].

While several authors and industry stakeholders have advocated for integrated management systems for the construction industry [14], there is no single integrated SHE management framework for construction organisations to use. Therefore, there is a general lack of a robust systematic mechanism that enables construction companies to ascertain the maturity of their SHE management practices. A process improvement tool, like a capability maturity model, can offer such a mechanism. Though maturity models have been proven valuable for assessing organizational processes or practices in delivering performance for various domains, there are just a few related examples of its application to integrated SHE management in the construction industry. For instance, Hamid et al. [15] developed the integrated management system for safety, health and environmental quality (SHEQ-MS) in the construction industry. Rebelo et al. [16] also developed the integrated management system-quality, environment and safety (IMS-QES). Though closely related to SHEM-CMM, these two systems/models do not enable SHE management capability maturity assessment. In addition to these two, there have been single stand-alone maturity models (i.e., maturity models that do not integrate multiples domains) developed for safety management [17] and environmental management [18], which still do not incorporate the environmental aspects and safety aspects, respectively.

Drawing on the afore-mentioned gap, the question that arises is what integrated safety, health and environmental capability maturity model can best work for the construction industry. To answer this question, this study thus examines integrated SHE management capability. It adopts the capability maturity modelling concept for the development of an integrated safety, health and environmental management capability maturity model (iSHEM-CMM) to enable the assessment of iSHEM capability of construction firms, effec-

tive management of components of an iSHEM system, and thereby improve construction iSHEM capability maturity levels and management practices. The capability maturity model developed is a useful assessment framework or tool for industry stakeholders, particularly construction firms, to evaluate the status of their current SHE management capability, identify strengths and improvement areas, and accordingly prioritise strategies/actions for improving their SHE management. Furthermore, clients who appoint construction companies could use the model as part of prequalification arrangements in selecting construction companies with an adequate SHE management capability.

The next section presents a brief overview of integrated SHE management capability and capability maturity modelling concepts to provide the foundation for developing the iSHEM-CMM. The research method applied, including the design decisions involved in developing the iSHEM-CMM, the maturity model and validation of the model, are subsequently presented. The implications stemming from the developed capability maturity model and concluding remarks are also presented.

## 2. Literature Review

This section conducts a critical comparative review of the related literature. Literature reviewed is presented under two sub-sections, i.e., safety, health and environmental management capability in construction; and capability maturity models. A systematic review through content analysis of literature related to the theme under investigation was conducted. Multiple queries were conducted on online databases like Google Scholar, Web of Science (WoS), Scopus and the like. Literature relevant to the current study and which spanned 1990 to 2019 were covered. Initial reviews limited to titles and abstracts of papers were accessed to ensure relevance to the theme under investigation. The search for the relevant literature was carried out using a combination of words like 'safety and health in the construction industry', health and safety management in construction', safety and health capability in construction', 'safety, health and environment', and 'capability maturity models'. Sections 2.1 and 2.2 summarise the significant literature retrieved from the search.

### 2.1. Safety, Health and Environmental Management Capability in Construction

The increasing concern regarding environmental, safety and health issues and their efficient management in construction have become of utmost importance for construction organisations worldwide. Some of these organisations are complying with SHE legislation and standards by deploying systematic and proactive initiatives such as SHE management systems [19–21] to address SHE issues and their associated undesirable outcomes. Though the adoption and implementation of SHE management systems are minimal in the construction sector, several studies have highlighted their implementation in construction as an innovative approach that offers substantial improvements in operation efficiency, standard compliance, as well as in SHE performance [22–24]. A study by Yoon et al. [24] revealed that in Korea the implementation of SHE standards saw safety performance increasing by more than 30%, with fatal accidents decreasing by 10.3%. Zeng et al. [22] reported that construction companies in China were able to enter international markets, reduce waste and noise control and improve safety and health at workplaces by implementing Environmental Management Systems (EMS). These studies show that the implementation of EMSs and SHMSs is an important innovative, systematic and proactive approach in reducing construction accidents and in minimising detrimental environmental impacts of construction operations [22,24]. However, the parallel implementation of both management systems (i.e., EMS and SHMS) have been criticised for being bureaucratic, costly, paperdriven and arduous [14,19,25], hence the need for an integrated management of SHE issues in construction through a single system (i.e., an integrated SHE management system).

Integrated SHE management in construction involves identifying, assessing and managing SHE risks rightly to minimise injuries, illness, fatalities and negative environmental impacts. It requires construction companies to take into account SHE considerations in addition to cost, time and quality considerations in all phases of building and construc-

tion projects. Though the compliance to SHE regulations often leads to a reduction in work-related tragedies and adverse environmental impacts, the efficient and effective management of SHE problems in construction based on an integrated SHE management framework makes it critical for construction companies to have an appropriate organisational capability which encompasses the policies, systems, resources, information, infrastructure and personnel of the company. However, empirical work into integrated SHE management capability is missing in the growing body of construction SHE management literature. For instance, within the last few decades, several studies on SHE management systems in the construction industry have focused on: (1) awareness, motivators, costs, benefits and barriers of management systems [2,26–28]; (2) effectiveness of SHE management systems in addressing occupational accidents, SHE performance, pollution and waste reduction [24,29]; (3) integration of environment, quality, safety and health management systems and benefits [14,19,25]; and the elements of stand-alone management systems and integrated management systems [29,30]. In terms of integrated SHE management capability, there is inadequate empirical research for insights into what constitutes integrated SHE management capability and mechanisms by which it can be reliably assessed to pave the way for continuous process improvement. Given the increasing concerns over SHE performance for sustainable construction and the lack of existing frameworks for integrated SHE management in construction, the development of a simple and implementable iSHEM framework that involves capabilities or practices relevant to the efficient implementation of an iSHEM system in a construction firm is crucial. The efficient management of iSHEM capabilities or practices could lead to better SHE performance. According to the capability maturity modelling concept, the degree of process effectiveness and efficiency reflects the capability of firms and organisations to implement processes successfully, thereby showing the maturity of organisational practices [31]. Therefore, capability maturity models (CMMs) serve as a good reference framework for developing iSHEM-CMM.

### 2.2. Capability Maturity Models

To respond to the highly competitive external environment, organisations continuously search for effective new approaches for assessing performance and organisational capability, as well as enhancing management capabilities [32] such as business excellence models, balanced scorecards, maturity models, total quality management, business process reengineering amongst others. However, amongst these approaches, maturity models have been designed to provide organisations with guidance on how to effectively measure and improve the maturity of functional domains within these organisations [33]. Moreover, they have assisted organisations in overcoming challenges of the need for cost reduction or quality improvement in the face of competitive pressure [34].

The principal idea of maturity models is that they describe the characteristics of organisational processes or activity at different levels of maturity [35]. They have their roots in quality management and continuous process improvement [36]. In particular, the Quality Management Maturity Grid by Crosby [37] describes the behaviour exhibited by a company at five maturity levels for a set of aspects of quality management [38]. According to Van Looy et al. [36], the best-known derivative of the quality management maturity concept is the capability maturity model (CMM). CMM was developed by the Software Engineering Institute (SEI) at Carnegie Mellon University as a reference model for assessing, evaluating and improving software process maturity [39,40]. Capability maturity models (CMMs) focus on improving organisational processes and identifying several levels of maturity ranging from low to high and each maturity level details the behaviour exhibited by organisations [41,42]. The CMM framework describes the maturity of organisations according to five levels (i.e., initial, repeatable, defined, managed and optimising) and determines these levels based on key process areas or capabilities [38,39]. However, the number of maturity levels can differ, depending on the domain and the concerns motivating the model.

The CMM Integration (CMMI), which is an extension of the CMM, is a single and comprehensive framework that is appropriate for organisations of any structure and focused on guiding organisation-wide process improvement [36,41]. It has two different representations of maturity, namely staged and continuous representations [43]. The staged presentation includes five levels similar to those of the original CMM. Each maturity level consists of several process areas that are specifically demarcated to that stage. Organisations get assessed against their process areas' existence or absence and produce an overall maturity level rating [44]. This presentation is useful for organisations that are looking at improving their overall process capability. With the continuous representation, maturity within each process area is analysed separately and improvements are made accordingly [36]. This presentation offers a more flexible approach to process improvements and is suitable for organisations looking to improve specific process areas and desiring to choose areas of implementation [45].

Currently, CMM/CMMI is one of the most widely accepted frameworks for assessing organisational capability in a domain as part of continuous process improvement [39,42]. Increasingly, CMMI has become a tool used to assess and improve organisational processes, systems, products, and competencies on the evolutionary path towards excellence and attaining desired outcomes [46]. Although originally developed for process improvement within the software industry, the CMM/CMMI represent a generic framework for continuous process improvement and hence has been applied in varied domains in several industrial sectors, including construction in areas such as supply chain management, risks management, disability management, change management, Building Information Modelling, and e-business [44,47–50]. In the area of SHE in construction, CMM frameworks have also been applied, although not specifically to integrated SHE management. For instance, there is the safety culture maturity model by Fleming [51] to access safety culture maturity, the AC2E performance matrix by Carillon Plc [52] to assess construction site safety management, the health and safety maturity model by Goggin [53] to assess the maturity of safety management practices of a given construction company at the organisational level, the Environmental management maturity model of construction programs by Bai et al. [54], and also the Design for occupational safety and health capability model (DfOSH) by Manu et al. [55] to access the DfOSH capability maturity of design firms in the construction sector. Other than Goggin's model, which focuses on safety and health management practices in construction, the extant literature does not reveal any other maturity models and systematic approaches for evaluating integrated SHE management in the construction industry, thus highlighting the significance of this paper. The application of CMMI/CMM in several areas in construction, including occupational safety, health and the environment, as a useful and robust tool for assessment and continuous process improvement, therefore, supports its application to integrated SHE management to produce an iSHEM-CMM.

## 3. Materials and Methods

In developing the iSHEM-CMM, the approach of Maier et al. [56] on how to develop maturity grids based on organisational capability assessments was as follows. Maier's et al. [56] procedural approach consists of four steps: (1) Planning: identification of target audience, aim, scope and success areas; (2) Development: defining the various parts of the maturity model which are the process areas, maturity levels, the cell descriptors and administration mechanisms; (3) Evaluation: model verification, refinement and validation; (4) Maintenance: documentation and communication of development processes, results and changes in process areas and cell descriptors. The main design decisions in this approach are the establishment of (1) key process areas (i.e., integrated SHE management capability attributes) and (2) the capability maturity levels.

In maturity model literature, maturity models (MMs) have received recurrent criticisms, particularly its lack of theoretical framework or methodology and traceability [57]. There is a dearth of literature on how to theoretically develop a maturity model [58].

However, the development process is not demonstrated in most of the documentation of maturity models and grids. Notwithstanding, recent studies have sought to introduce a structured approach to previous work done [59].

Compared to other traditional methods, Maier et al.'s [56] was followed in this study because it provides rigorous and consistent development procedure, and also looks similar to some of the common steps in the approaches developed by other authors like SEI [60], De Bruin et al. [58], Poghosyan et al. [61], and Asah-Kissiedu et al. [12]. Sub-Section 3.1 expounds the various methods.

### 3.1. Maturity Model Development

Like most existing CMMI- or CMM-based models, the iSHEM-CMM follows the continuous-structure [60] since it provides a generic measurement of capability maturity level for each integrated SHE management capability attribute. The model is represented in a grid format and has two main components: capability maturity levels and integrated iSHEM capability attributes. Levels of capability maturity are allocated against the attributes, thereby creating a series of cells. Each cell contains a brief text description (i.e., descriptor) for each activity at each capability maturity level. The following subsections present the steps taken to develop the maturity model.

#### 3.1.1. Design Decisions for Developing the iSHEM-CMM

In this section, the main design decisions outlined in Maier et al. [56] for developing maturity grids are elaborated.

##### Planning

Step 1: Specifying the Audience. The iSHEM-CMM is intended to assist construction firms to improve their SHE management. The expected audience of the model is thus construction firms.

Step 2: Defining the Aim. The purpose of the iSHEM-CMM is to assist construction companies to improve SHE performance in the construction sector. The aim of the maturity model is, therefore, to assist construction firms to assess their current SHE management maturity to facilitate continuous improvement.

Step 3: Clarifying the Scope. While some maturity models are designed for generic purposes, others are designed for a specific domain. The iSHEM-CMM, as the name indicates, is designed to support a particular domain, which is SHE management in the construction industry.

Step 4: Defining the Success Criteria. The development of the iSHEM capability maturity model is motivated by the need for improved guidance on SHE management processes and practices in the construction industry. The most important success criteria were, therefore: (1) Usefulness for the construction industry, determined by the relevance of the domain's components, and the ability of the model to support improvement effort within SHE management; (2) Usability determined by the clarity and the syntactic quality of the model; and (3) Coverage of key iSHEM capability attributes determined by how well the maturity model covers the areas important to focus on for ensuring effective management of SHE issues in construction companies.

##### Development

Step 5: Selecting the Process Areas. A key element of developing a maturity model is the identification of capability areas/attributes [56,58,61]. Therefore, the development of the iSHEM-CMM involved identifying the relevant key iSHEM capability attributes and the definitions of the levels of maturity. According to Maier et al. [56], the key process areas used in developing a maturity grid can be derived from (1) the experiences in the field of the originator and by reference to established knowledge in a particular domain; and (2) a panel of experts in the domain, especially where there is limited prior literature concerning the domain.

Considering the lack of empirical work on construction SHE management capability, this study used a panel of construction industry experts after a comprehensive systematic literature review to identify potential capability attributes for achieving effective integrated SHE management in construction. This was applied as a three-pronged sequential research approach comprising: (1) a systematic literature review to identify potential integrated SHE management capability attributes and a preliminary expert verification process to ascertain the appropriateness and comprehensiveness of the identified attributes; (2) application of expert Delphi technique to generate consensus regarding the importance of the attributes; and (3) application of voting analytical hierarchy process (VAHP) to generate weights of importance based on the outcomes of the Delphi technique. Detailed description of the application of the three-pronged approach is given in Asah-Kissiedu et al. [12].

To select suitable and qualified experts for the preliminary verification of capability attributes, the guidance suggested by Hallowell and Gambatese [62] in selecting experts was followed. This included at least five years of professional experience in the construction industry, a minimum of five years' experience in SHE management, an advanced degree in construction management or other related fields (minimum of BSc.), an affiliation with a professional body and have researched areas of environmental, health and safety management in construction. In line with the criteria, twelve (12) experts were engaged for the preliminary verification process and a total of 30 experts for the three-round Delphi survey. The experts for the preliminary verification were academics with industry experience and expertise in SHE management in construction. Such people are likely to have an up to date understanding of the subject matter of the study. Therefore, they were considered useful to engage with in the preliminary verification exercise prior to the Delphi survey. The experts involved in the Delphi survey were industry professionals.

Each of the Delphi rounds took three weeks, spanning a three-month duration. From the verification process and the Delphi rounds, the views of the experts regarding the capability attributes were collated and analysed. An agreement on 20 iSHEM capability attributes was then obtained (refer to Section 4). A detailed account of the derivation of the attributes from the aforementioned methods is reported in Asah-Kissiedu et al. [12]

Step 6: Formulating the Maturity Levels and Descriptors. The literature shows that CMMs commonly used five maturity levels [56,61,63], which aligns with the original CMM by Paulk et al. [39] Similarly, in this study, five capability maturity levels (i.e., Level 1 being the lowest maturity level and Level 5 being the highest maturity) were adopted as shown in Table 1. Capability maturity level definitions and characteristics were abstracted from the literature review and refined through expert review. In line with the guidelines by Maier et al. [56], the maturity level descriptors at the extreme ends (i.e., Level 1 being the lowest maturity level, and Level 5, being the highest maturity) were formulated based on the underlying notion of what represents maturity for each attribute. In capability maturity modelling, lower levels of maturity are used as the basis for achieving higher levels of maturity. For instance, for a construction firm to reach capability Level 5 or full maturation in a capability attribute, it should have met the requirements for the lower levels. As a result, each level is defined and characterised clearly, thus allowing companies to self-evaluate their level of maturity. It is therefore important to understand what these capability maturity levels represent in practice, as they are fundamental to assessing the capability maturity of a company. Shown below in Table 1 are the capability maturity levels and their definitions.

Step 7: Formulating the Cell Texts (i.e., maturity level descriptors). This decision point represents the intersection of the key process area (i.e., the capability attributes) and the capability maturity levels. Attribute characteristics, thus, need to be described at each level of maturity. This decision point is recognised as a significant step in developing a maturity model assessment [56]. To be able to formulate cell descriptors that are precise, concise, and clear, three considerations are described by Maier et al. [56]: (1) using a top-down or bottom approach; (2) consideration of the information source; (3) consideration of the formulation mechanism. The top-down approach involves the writing of definitions

before measures or a set of practices are developed to fit the definitions, while the bottom approach involves the determination of measures before definitions are written to reflect the measures [56]. Since integrated SHE management in construction is a relatively new field in maturity model applications, not much evidence is available for what is thought to represent maturity. Consequently, a top-down approach was deemed appropriate for formulating the cell texts since this approach places emphasis first on what maturity is before how it can be measured [56]. Again, this approach was used because of the lack of empirical work on integrated SHE management capability.

**Table 1.** Capability maturity levels and definitions.

| Capability Level | Definition |
| --- | --- |
| Level 1 | There are no structured processes and procedures in place. Performance is consistently poor. |
| Level 2 | Organisational processes and procedures may exist but are usually ad-hoc and unstructured. Procedures and processes are not defined. Performance is fair. |
| Level 3 | Organisational processes and procedures are formal and defined. Process and procedure are reactive. Performance is mostly good. |
| Level 4 | Organisational procedures and processes are planned, well-defined, proactive and generally conform to best practices. Performance is very good and consistently repeated. |
| Level 5 | Organisational processes and procedures are standardised, fully integrated throughout the organisation, and continually monitored, reviewed for continuous improvement. Performance is exemplary and comparable to best in the industry. |

In establishing what represents maturity in each of the key process areas (i.e., SHE management capability attribute) in this study, the underlying notion of maturity was obtained by reviewing various sources, including extant literature relating to the key process areas, feedback from future recipients of the model (through an expert verification), existing capability maturity models and best practice guides on subjects related to SHE management capability attributes. Therefore, existing capability maturity models like the UK Coal Journey Model by Foster and Hoult [17] and Risk Management Maturity Model (RM3) by the Office of Road and Rail, and Health and Safety Maturity [63] were reviewed to obtain the underlying notion of maturity for each of the SHE capability attributes. In summary, the cell texts were formulated using:(1) The underlying rationale of maturity of each capability attributes; and (2) The identification and the descriptions of the best and worst practices at the extreme ends of the scale (i.e., Level 1 and Level 5), such that Level 1 represented no or very low maturity and Level 5 represented the highest level of maturity which is also presented by reviews within the capability maturity model literature to ensure continuous improvement. Secondly, the other cell descriptors in between (i.e., levels 2, 3 and 4) were also deduced from the underlying notion and formulated accordingly. In the end, the model was developed with a fraction full version presented in the results section and the full version in the Appendix A, Table A1.

Step 8: Defining the Administration Mechanisms. The iSHEM-CMM was developed as a stand-alone model and targeted for application in several construction firms. Following the formulation of cell texts, the developed model and an evaluation questionnaire were sent to selected experts to further verify the model.

Evaluation

Step 9: Validating the Model. Once the iSHEM-CMM was populated, it was evaluated by construction professionals to ensure the practical utility of the model. A detailed description of the model validation is presented in Section 5.2.

Maintenance

Step 10: Documenting, communicating and maintaining the model. The purpose of the maintenance phase is to keep the final maturity model and, its elements or attributes

current. Continued accuracy and relevance of the model can be ensured by its end-users during this phase. For the iSHEM-CMM, communication is in part secured through this paper.

## 4. Results and Discussion

The experts for the preliminary verification and Delphi survey were experts who had knowledge and experience in SHE management in the construction industry. Each of the experts is affiliated with at least one professional body, which includes: Chartered Institute of Building, Institute of Environmental Management and Assessment, Institution of Occupational Safety and Health, International Institute of Risk and Safety Management, Association of Project management, Ghana Institution of Construction, Ghana Institute of Safety and Environmental Professionals and Ghana Institute of Surveyors. The years of experience in SHE management in construction are between 5 and 17 years. The experts engaged in the study were suitable as their experience and roles relate to SHE management in construction.

From a systematic literature review, twenty-seven (27) potential capability attributes were identified. At the end of the verification process and the subsequent three-round Delphi survey, 20 integrated SHE management capability attributes were finally obtained and subsequently categorised, based on their relatedness, into five thematic areas of SHE management capability. The five thematic categories are strategy, people, process, resources, and information. The categorisation of the capability attributes is consistent with the concept of organisational capability maturity, although specific to integrated SHE management [39,64–66]. Detailed descriptions of the thematic categories and the various attributes within are presented in Table 2. The emergent iSHEM capability attributes were similar to some of the key process areas/capabilities/criteria used in existing capability maturity grids and models. For example, Goggin's [53] Health and safety maturity model for health and safety in construction proposed attributes such as management commitment, safety policy, hazard identification, resources, reporting and control, and worker involvement and commitment. The UK coal journey maturity model by Foster and Hoult [17] also included attributes such as policy and commitment, training and competence, communication and consultation, documents and operations control, incident investigations, and monitoring and auditing.

Furthermore, the design safety capability maturity model for the offshore sector by Strutt et al. [46] included attributes like education and training, research and development, organisational learning, and managing of safety in the supply chain, while the safety culture maturity model by HSE [67] also included attributes including 'training', 'management commitment and visibility', 'learning organisation', and 'safety resources'. Furthermore, the design for occupational safety and health (DfOSH) maturity grid by Manu et al. [66] proposed attributes such as DfOSH competence and training, management commitment, design risks management, physical work and ICT resources. In the building information modelling (BIM) domain in construction, Succar [68] proposed a BIM maturity matrix, which was comprised of capability attributes such as leadership, physical infrastructure, technology (encompassing software and hardware) and human resources (comprising of knowledge, resources and skills). The iSHEM capability attributes (e.g., senior management commitment to SHE; SHE policy objectives and targets; SHE management programme; SHE risk management; Management of outsourced SHE services; Physical and financial resources; SHE incidents investigation; SHE system auditing; SHE training and processes for learning lessons and knowledge management) share similarities with the above-mentioned attributes in models by HSE [67], Strutt et al. [46], Goggin's [53], Filho et al. [69], Foster and Hoult [17] and Poghosyan et al. [61], although the iSHEM capabilities have specific relevance or focus on the implementation of an iSHEM system in construction firms.

**Table 2.** Verified integrated SHE management capability attributes.

| Thematic Category | Attributes |
|---|---|
| Strategy, i.e., the organisation's vision and top management commitment to SHE management | Senior management commitment to safety, health and environment (SHE) management |
| | An integrated SHE policy that serves as the foundation for a company's SHE development and implementation |
| | SHE objectives and targets for a company, in line with SHE policy |
| | SHE management programme, i.e., company's action plans for achieving SHE objectives and targets |
| Processes, i.e., the organisation's procedures, processes and systems for SHE management | SHE risks management, i.e., systems, processes and procedures for SHE hazards identification, risks assessment and identification risks control strategies |
| | Management of outsourced services, i.e., processes and mechanisms for assessing the competence of outsourced personnel, subcontractors and suppliers with regards to the management of SHE |
| | SHE operational control, i.e., processes, procedures and measures for controlling SHE risks, to ensure SHE regulatory compliance in operational functions and to achieve the overall SHE objectives |
| | SHE emergency preparedness and responses, i.e., emergency procedures and measures to minimise the impact of uncontrolled events and unexpected incidents |
| | SHE performance monitoring and measurement, i.e., systems, processes and procedures to monitor and measure SHE performance to ensure compliance with SHE regulations |
| | SHE incidents investigation, i.e., processes and procedures for investigating the causes of SHE incidents |
| | SHE system auditing, i.e., processes and procedures to conduct SHE audits to assess compliance and SHE management system effectiveness |
| People, i.e., organisation's human capital, their roles, responsibilities, and involvement in SHE management | SHE roles and responsibilities, i.e., availability of dedicated SHE roles and responsibilities within an organisational hierarchy |
| | SHE Training, i.e., provision of suitable SHE training for personnel |
| | Employee involvement and consultation at all levels in SHE management and operations |
| | SHE competence, i.e., the skills, knowledge and experience of personnel to undertake responsibilities and perform SHE activities |
| Resources, i.e., organisation's physical and financial resources required for SHE management | Physical SHE resources, i.e., provision of physical resources for SHE implementation |
| | Financial resources for SHE, i.e., Provision of financial resources for SHE implementation |
| Information, i.e., SHE related documents, data, lessons, records and their communication across an organisation | Communications, i.e., communication of relevant SHE information and requirements to personnel and other relevant stakeholders |
| | SHE documentation and control, i.e., provision and maintenance of adequate SHE documentation and records |
| | SHE lessons and knowledge management, i.e., capturing lessons learned and knowledge acquired from historical incidents and management of SHE |
| | Communications, i.e., Communication of relevant SHE information and requirements to personnel and other relevant stakeholders |

### 4.1. The iSHEM Capability Maturity Model

After the capability attributes and the capability, maturity levels were obtained and an initial iSHEM-CMM was developed. The model is a multilevel framework and offers maturity assessment on a five-level scale, within five thematic categories consisting of 20 integrated SHE management capability attributes. Table 3 shows an excerpt of the iSHEM

capability maturity grid, with two iSHEM capability attributes—Senior management commitment to SHE and SHE policy and maturity levels from 1–5. Due to its large size, the full version of the iSHEM-CMM is presented in the Appendix A, Table A1 of this paper.

### 4.2. Validating the iSHEM Capability Maturity Model

In this section, validation of the maturity model by industry experts is presented. De Brium et al. [58] recommended that the evaluation process of a maturity model should mainly focus on the model's constructs (i.e., relevance and coverage of the domains components) and the model instruments (i.e., the reference model, performance scale and assessments procedure). In view of this, the validation process involved: (1) the use of the iSHEM by construction professionals to assess the SHE management capability of construction companies; and subsequently (2) the completion of a validation survey by the professionals. The validation survey was used to appraise both content of the maturity model (i.e., the relevance and appropriateness of the capability attributes and levels) and its usability (i.e., understandability, ease of use and practicality).

#### 4.2.1. Selection of Companies for Validation

To ensure a broad validation of the maturity model, construction professionals, including SHE experts from 70 construction firms operating in Ghana, were invited to participate in the validation process. Fifty-nine (59) construction firms consented. The construction professionals involved included Health and Safety managers (15.3%), Project managers and construction managers (13.6%), Environmental Managers (13.6%), and Site Managers, Safety, Health or Environmental Consultants and Health and Safety Officers (11.9%). A majority of the respondents (67.8%) had over five years of professional experience. This is indicative of an experienced and knowledgeable group of construction professionals.

#### 4.2.2. Questionnaire for Validation

To validate the capability maturity model (i.e., SHEM-CMM), an evaluation questionnaire was used as the instrument (see Appendix A-Table A2). It consisted of two sections. The first section solicited information on the respondent background details. In the second section, respondents were asked to evaluate the model based on six criteria (i.e., relevance of attributes, comprehensiveness of attributes, appropriateness, adequacy of capability maturity levels, ease of understanding, ease of use and level of usefulness and practicality). These validation criteria were similar to the survey developed by Salah et al. [70] (2014).

The validation exercise required that the construction professionals were to assess their company's SHE management capability maturity by using the developed maturity model and thereafter evaluate the capability maturity model as a whole by the six criteria on the five-point Likert scale using the levels (5) Strongly agree, (4) agree, (3) Neither agree nor disagree, (2) disagree, (1) Strongly disagree.

#### 4.2.3. Validation Results

In total, responses were obtained from 59 construction firms operating in Ghana. The results of the survey are presented in Table 4. From the validation results, it is evident that the iSHEM-CMM is comprehensive and suitable for assessing the SHE management capability maturity of construction companies. The high rating indicates a convincing level of approval of the developed capability maturity model. Regarding the relevance and comprehensiveness of integrated SHE management capability attributes, the results confirm that the capability attributes are relevant and did cover all aspects of integrated SHE management capability in construction. Concerning the correct assignment of attributes to their respective capability levels and sufficient maturation of attributes, the validation results indicated that the construction professionals were satisfied with the accuracy of the capability attributes and its correct assignment to their respective maturity levels in the developed model.

**Table 3.** The iSHEM-CMM (excerpt).

| She Capability Attributes | Underlying Notion of Maturity | Capability Maturity Levels | | | | |
|---|---|---|---|---|---|---|
| | | Level 1 | Level 2 | Level 3 | Level 4 | Level 5 |
| Senior management Commitment | As maturity increases, senior management commitment to safety, health and environmental (SHE) management becomes unwavering, visible and well-articulated across the company | • Lack of senior management commitment to SHE management<br>• There is no resource commitment (financial and human resources) for SHE related issues | • Limited commitment by company's senior management to SHE implementation<br>• Limited resource commitment for SHE related issues | • Partial commitment by company's senior management to SHE implementation<br>• Show of senior management commitment is reactive (e.g., when significant risks are anticipated or response to a major environmental impacts)<br>• An ad hoc implementation committee is established<br>• SHE champion is identified<br>• There is resources commitment for SHE-related issues | • Firm commitment by company's senior management to SHE implementation.<br>• Senior management commitment is aligned to company's policy on SHE management.<br>• Senior management are amongst the SHE champions within the organisation.<br>• Management commitment is well articulated across the company<br>• Sufficient resources commitment for SHE-related issues | • There is a full, unwavering and clearly visible commitment of company's senior management to SHE implementation<br>• Senior management continuously and visibly demonstrate their commitment to SHE and show shared values directed at continually meeting SHE objectives safely<br>• A cross functional SHE implementation committee is established including a SHE champions and members from all key management functions of the company.<br>• There is a ring-fenced resource commitment for SHE implementation and maintenance<br>• Company senior manager(s) are amongst SHE management champions within the industry and are recognised as industry thought-leaders in respect of SHE management |

**Table 3.** *Cont.*

| She Capability Attributes | Underlying Notion of Maturity | Capability Maturity Levels | | | | |
|---|---|---|---|---|---|---|
| | | **Level 1** | **Level 2** | **Level 3** | **Level 4** | **Level 5** |
| She Policy | As maturity increases, company SHE policy becomes explicitly stated, well-communicated within the organisation, and interpreted and applied consistently by all managers/supervisors and staff. | • No policy statement on SHE management | • SHE policy statement is outdated and vaguely worded<br>• SHE policy does not meet legal requirements and employees are rarely involved in its development<br>• Policy has not been communicated within the company and documented | • SHE policy statement is clear, setting out the intention(s) on how SHE is managed, tracked and reported<br>• Policy meets majority of legal requirement with some employees actively involved in its development<br>• Policy is communicated across different levels of the company, but management or supervisors and employees have inconsistent interpretations and applications of the policy<br>• 3Policy statements are poorly documented and not displayed at workplace | • SHE policy is clear, comprehensive and well-defined, setting out the intention on SHE<br>• SHE policy presents a clear approach to managing SHE including the required accountability and responsibility for managing SHE<br>• SHE policy meets all the legal requirements and other requirements the company subscribes to<br>• More relevant employees are actively involved in SHE policy formation and strategy formulation<br>• SHE policy is actively communicated within the company and to other stakeholders<br>• Policy is accepted, understood and consistently interpreted and applied in the same way by all manager's or supervisors and employees<br>• SHE policy is formally documented, displayed at the workplace and is available to all stakeholders | • There is a clear policy on SHE management, setting out intention(s) on SHE management and recognising that SHE implementation is not a separate task but an integral part of the organisation SHE activities<br>• All relevant people are engaged in SHE policy formation as wells as SHE strategy formulation, with clear actions, and accountabilities and targets<br>• Documented policy is in place, consistent with other best-performing organisation's policies, communicated and readily available to all stakeholders<br>• SHE policy is periodically reviewed to ensure that it remains relevant to the company, reflect industry best practices and demonstrate effectiveness and continuous improvement |

**Table 4.** Summary of responses feedback for maturity model evaluation.

| Assessment Criteria | Evaluation Response (%) ($n$ = 59) | | | | | | |
|---|---|---|---|---|---|---|---|
| | Strongly Agree | Agree | Neither Agree nor Disagree | Disagree | Strongly Disagree | Total (%) | Median/Mean/Standard Deviation |
| **Attributes used in the SHEM-CMM worksheet** | | | | | | | |
| Attributes are relevant to SHE management capability. | 35.6 | 62.7 | 1.7 | 0 | 0 | 100 | 4.00/4.34/0.51 |
| Attributes cover all aspects of SHE management capability. | 20.3 | 62.7 | 16.9 | 0 | 0 | 100 | 4.00/4.03/0.62 |
| Attributes are correctly assigned to their respective capability level. | 15.6 | 71.2 | 13.6 | 0 | 0 | 100 | 4.00/4.02/0.54 |
| Attributes are clearly distinct. | 40.7 | 50.8 | 8.5 | 0 | 0 | 100 | 4.00/4.32/0.63 |
| **Capability maturity levels** | | | | | | | |
| The capability levels sufficiently represent maturation in the attributes. | 18.6 | 69.5 | 8.5 | 3.4 | 0 | 100 | 4.00/4.03/0.64 |
| There is no overlap detected between descriptions of maturity levels. | 6.8 | 52.5 | 27.1 | 13.6 | 0 | 100 | 4.00/3.53/0.82 |
| **Ease of understanding** | | | | | | | |
| The capability levels are understandable | 33.9 | 61 | 5.1 | 0 | 0 | 100 | 4.00/4.29/0.56 |
| The documentations (i.e., assessment instructions) are easy to understand | 13.6 | 71.2 | 11.9 | 3.4 | 0 | 100 | 4.00/3.95/0.63 |
| The results are understandable | 13.6 | 79.7 | 6.8 | 0 | 0 | 100 | 4.00/4.07/0.45 |
| **Ease of use** | | | | | | | |
| The scoring scheme [i.e., drop-down options for maturity levels (1–5)] is easy to use | 39 | 57.6 | 1.7 | 1.7 | 0 | 100 | 4.00/4.39/0.61 |
| The SHEM-CMM is easy to use | 18.6 | 71.2 | 8.5 | 1.7 | 0 | 100 | 4.00/4.07/0.58 |
| **Usefulness sand practicality** | | | | | | | |
| SHEM-CMM is useful for assessing SHE management capability | 49.2 | 47.5 | 3.4 | 0 | 0 | 100 | 4.00/4.46/0.57 |
| SHEM-CMM is practical for use in industry | 28.8 | 64.4 | 6.8 | 0 | 0 | 100 | 4.00/4.22/0.56 |

Furthermore, the results indicated that the majority of the construction professionals were of the opinion that capability levels, supporting documentations and the results were easy to understand. Additionally, the iSHEM-CMM was found to be easy to use, useful for assessing SHE management capability and practical for use in the construction industry by the majority of the construction professionals, particularly the ease of using the Microsoft Excel format of the maturity model and the user-friendly nature of the scoring scheme (i.e., drop-down options for capability levels) during the assessment. Based on the overall results of the validation exercise, the developed integrated SHEM-CMM was generally well-received by practitioners in the industry.

## 5. Conclusions

Efficient management of SHE issues has become of utmost importance for construction firms globally. Construction firms need to have the appropriate capability in terms of SHE to effectively minimise injuries, illness and negative environmental impacts through an integrated SHE management framework. Construction firms would have differing iSHEM capabilities, and it is important they understand their current iSHEM capability depth so that they can continuously improve. Similarly, it is vital that construction clients, consultants or other institutions engaging the services of construction firms are also able to evaluate the iSHEM capability of those organisations. This study adopted the capability maturity modelling concept to develop an integrated safety, health and environmental management capability maturity model for construction firms. This study addressed a significant research gap relating to iSHEM capability by identifying 20 distinct capability attributes and presenting empirical work on developing an iSHEM capability maturity model to facilitate assessments and improvement of integrated SHE management practices. The maturity model shows five maturation levels in distinct iSHEM capability attributes drawn from literature review and a Delphi survey with a panel of SHE experts. To ensure the usefulness of the maturity model in practice, the model was further improved by SHE expert verification and refinement and validated by construction professionals, including SHE experts. The findings reveal that the developed model is fit and is capable of assessing the iSHEM capability of construction firms, managing components of an iSHEM system effectively, and improving construction iSHEM capability maturity levels and management practices. The developed maturity model considers the main tasks of an iSHEM system, the underlying processes, strategies, resources, information and the people using the iSHEM system. It is anticipated that the developed iSHEM-CMM would be beneficial to construction firms and other industry stakeholders by undertaking self-assessment of their iSHEM capability to better understand their iSHEM and implement actions needed to improve it.

### 5.1. Implications of the Research

This study's main implication is that the developed model provides a means for construction companies to evaluate their SHE management capability systematically. This would enable them to ascertain the areas of strength and deficiency in respect of their SHE capability. On the basis of SHE management capability self-assessment, construction companies could prioritise their investments and target efforts at addressing any identified areas of capability deficiency to ensure continuous improvements and avoid sub-optimisation. Moreover, this model serves as a management tool that allows SHE management consultants to evaluate their construction clients firm's current SHE capability maturity and provide guidance on how they can improve their SHE management practices and processes. Additionally, the identified iSHEM capability attributes could be used by construction clients (including government agencies) as part of the SHE management criteria for selecting companies to undertake projects.

Several maturity models have been developed in relation to safety culture in many domains; however, until now, no maturity model has been developed that has attempted to take the maturity perspective into integrated SHE management practices in construc-

tion. The output of this study (i.e., the iSHEM-CMM) therefore offers a basis for similar capability maturity-based research with a focus on integrated SHE management capability in other industries.

### 5.2. Limitations and Recommendation for Further Studies

The research has limitations which are highlighted in this section. The study was based on professional views of SHE management experts and other practitioners within the Ghanaian construction industry; therefore, findings may be peculiar to SHE management in the Ghanaian construction industry. Further studies could be conducted among SHE management experts and practitioners in the construction industry of other countries to enable an appropriate comparison to be done. Another limitation identified lies in the fact that the development of the integrated SHEM-CMM focused on the construction industry alone. This may tamper with its immediate applicability to other industrial sectors. Future studies could be conducted to develop a similar model for other industries other than the construction industry. This can improve the SHEM of such industries as well.

Furthermore, another potential limitation relates to the sample size used to validate the maturity model. Available guidance for testing CMM using an expert evaluation approach [70] does not specify the minimum number of experts. Nonetheless, for expert group techniques, the recommended number of experts range from 8–12 (e.g., Delphi Technique [66]). This is because in an expert group technique, the focus tends to be on the depth of knowledge of the experts rather than the breadth of participation, i.e., the number of experts [66]. Therefore, in this study, the number of experts that were involved in the CMM can be deemed to be adequate. Notwithstanding, future studies could adopt alternative methods, e.g., large cross-sectional surveys to test the capability model.

**Author Contributions:** Conceptualization—M.A.-K., P.M., C.A.B. and A.-M.M.; methodology—M.A.-K., P.M., C.A.B. and A.-M.M.; validation—M.A.-K., P.M., C.A.B. and A.-M.M.; formal analysis—M.A.-K.; investigation—M.A.-K.; writing—original draft preparation—M.A.-K. and K.A.; writing—review and editing—M.A.-K., K.A., P.M., C.A.B. and A.-M.M.; supervision—P.M., C.A.B. and A.-M.M.; project administration—M.A.-K., P.M., C.A.B. and A.-M.M.; funding acquisition—M.A.-K. and P.M. All authors have read and agreed to the published version of the manuscript.

**Funding:** Appreciation is extended to the Commonwealth Scholarship Commission (Reference No: GHCS-2016-147) for funding this research.

**Institutional Review Board Statement:** This study was granted ethical approval by the research ethics committee of the Faculty of Environment and Technology, University of the West of England (UWE), Bristol, UK, (reference # FET.17.11.013; date of approval 12 January 20218). Further information about UWE Bristol's research governance and policies is available at https://www.uwe.ac.uk/research/policies-and-standards, accessed on 4 October 2021.

**Informed Consent Statement:** Informed consent was obtained from all subjects involved in the study.

**Data Availability Statement:** All data are a part of the manuscript.

**Conflicts of Interest:** The authors declare no conflict of interest.

# Appendix A

**Table A1.** Integrated safety, health and environmental management capability maturity model (iSHEM-CMM).

| | Integrated Safety, Health and Environmental Management Capability Maturity Model (iSHEM-CMM) | | | | |
|---|---|---|---|---|---|
| **She Capability Attributes** | **Capability Levels** | | | | |
| | **Level 1** | **Level 2** | **Level 3** | **Level 4** | **Level 5** |
| Senior management Commitment | • Lack of senior management commitment to SHE management<br>• There is no resource commitment (financial and human resources) for SHE related issues | • Limited commitment by company's senior management to SHE implementation<br>• Limited resource commitment for SHE related issues | • Partial commitment by company's senior management to SHE implementation<br>• Show of senior management commitment is reactive (e.g., when significant risks are anticipated or response to a major environmental impacts)<br>• An ad hoc implementation committee is established<br>• SHE champion is identified<br>• There is resources commitment for SHE related issues | • Firm commitment by company's senior management to SHE implementation<br>• Senior management commitment is aligned to company's policy on SHE management<br>• Senior management are amongst the SHE champions within the organisation<br>• Management commitment is well articulated across the company<br>• Sufficient resources commitment for SHE-related issues | • There is a full, unwavering and clearly visible commitment of company's senior management to SHE implementation<br>• Senior management continuously and visibly demonstrates their commitment to SHE and show shared values directed at continually meeting SHE objectives safely<br>• A cross functional SHE implementation committee is established including a SHE champions and members from all key management functions of the company<br>• There is a ring-fenced resource commitment for SHE implementation and maintenance<br>• Company senior manager(s) are amongst SHE management champions within the industry and are recognised as industry thought-leaders in respect of SHE management |
| She Policy | • No policy statement on SHE management | • SHE policy statement is outdated and vaguely worded<br>• SHE policy does not meet legal requirements and employees are rarely involved in its development<br>• Policy has not been communicated within the company and documented | • SHE policy statement is clear, setting out the intention(s) on how SHE is managed, tracked and reported.<br>• Policy meets majority of legal requirement with some employees actively involved in its development<br>• Policy is communicated across different levels of the company, but management or supervisors and employees have inconsistent interpretations and applications of the policy.<br>• 3Policy statements are poorly documented and not displayed at workplace | • SHE policy is clear, comprehensive and well-defined, setting out the intention on SHE<br>• SHE policy presents a clear approach to managing SHE including the required accountability and responsibility for managing SHE<br>• SHE policy meets all the legal requirements and other requirements the company subscribes to<br>• More relevant employees are actively involved in SHE policy formation and strategy formulation<br>• SHE policy is actively communicated within the company and to other stakeholders<br>• Policy is accepted, understood and consistently interpreted and applied in the same way by all managers or supervisors and employees<br>• SHE policy is formally documented, displayed at the workplace and is available to all stakeholders | • There is a clear policy on SHE management, setting out intention(s) on SHE management and recognising that SHE implementation is not a separate task but an integral part of the organisation SHE activities<br>• All relevant people are engaged in SHE policy formation as well as SHE strategy formulation, with clear actions, and accountabilities and targets<br>• Documented policy is in place, consistent with other best-performing organisation's policies and is communicated and readily available to all stakeholders<br>• SHE policy is periodically reviewed to ensure that it remains relevant to the company, reflects industry best practices and demonstrates effectiveness and continuous improvement |

**Table A1.** *Cont.*

| Integrated Safety, Health and Environmental Management Capability Maturity Model (iSHEM-CMM) |
|---|

| | | | | |
|---|---|---|---|---|
| **She Objectives and Targets** | • No formal SHE objectives and targets identified and documented | • SHE objectives and targets are vaguely worded and not based on any baseline review of the company's SHE operations. They are not 'specific, measurable, attainable, relevant and timely (SMART) and prioritised.<br>• People in relevant functional area(s)are not involved in setting SHE objectives and targets<br>• Objectives and targets not included in critical tasks or role descriptions of employees<br>• SHE objectives and targets are poorly documented and not communicated to employees and other stakeholders | • SHE objectives and targets are defined, formal, based on a baseline review and consistent with SHE policy and applicable legal and other regulatory requirements<br>• Some SHE objectives and targets may be SMART and prioritised.<br>• Some people in relevant functional areas(s) are involved in setting objectives and targets<br>• Objectives and targets are rarely included role descriptions of employees<br>• SHE objectives and targets are somewhat documented and informally communicated to employees and relevant stakeholders within the company | • SHE objectives and targets are formal, well defined, mostly SMART, and consistent with SHE policy and applicable legal and other regulatory requirements<br>• More people in relevant functional areas (s)are involved in setting SHE objectives and targets<br>• Objectives and targets are included role descriptions of employees<br>• Objectives and targets are properly documented and formally communicated to all relevant functions across the company | • SHE objectives and targets are clear, SMART, prioritised and aligned to the overall SHE policy and focused towards continually improving SHE performance.<br>• All relevant people are involved in setting SHE objectives and targets<br>• Objectives and target are included in critical tasks or role descriptions of employees<br>• SHE objectives and targets are adequately documented, monitored, routinely reviewed and updated<br>• to ensure continuous improvement. |
| **She Management Programme** | • There are no clearer or well defined SHE management programme(s) for achieving objectives and targets. | • SHE plans and programme(s) are available but without a clear definition of specific responsibilities and the time frame.<br>• Little involvement of employees in establishing SHE plans and programme(s) | • Formal and detailed management plans and programme(s) are available<br>• Key responsibilities, tactical steps, resources needed and schedules are clearly defined to achieve SHE objectives and targets<br>• More involvement of employees in establishing SHE programmes | • SHE management plans and programme(s) are adequate, more detailed and integrated with company objectives, strategies and budgets<br>• Greater number of employees' involvement in establishing SHE programmes<br>• SHE plans and programme(s) are clearly communicated to all who need to know | • SHE management plans and programmes are dynamic and integrated with company's SHE planning strategies<br>• Full involvement of employees and other stakeholders in establishing SHE programmes<br>• SHE management programmes are continuously reviewed and modified to address changes to company's operations for continuous improvement of SHE programmes |
| **She Risk Management** | • No processes and procedures for SHE hazards identification, risk assessment and control | • Informal processes and procedures for SHE hazards identification and risk assessments are in place<br>• Risk control measures are poorly defined, understood and have limited application<br>• SHE risks assessments and management are poorly documented | • Formal processes and procedures for SHE hazards identification and risk assessment are in place<br>• Processes and procedures for identification and management of SHE risks, focuses on the most significant and obvious SHE risks<br>• SHE risks assessments are carried out in isolation<br>• Risk control measures are somewhat defined and used to reactively managed identified SHE risks<br>• Most important SHE risks assessment activities and plans are documented | • Formal, more detailed and proactive processes and procedures for SHE hazards identification and risk assessment<br>• Processes and procedures for identification and management focusses on specific, hazards and risks, including less obvious and immediate risks<br>• Processes and procedures are consistently applied to identify and manage SHE risks<br>• SHE risks control measures are well defined, understood and implemented in a consistent manner<br>• All levels of SHE employees and other stakeholders can contribute to risks assessments<br>• Appropriate SHE risks assessment records are accurately documented and maintained | • Well-defined processes and procedures for SHE risks management are in place and practicable<br>• SHE risk management processes and procedures are embedded into company's SHE planning activities and considered as a core measure of operational excellence<br>• The approach to SHE risks assessment are routinely applied consistently throughout the company in a pragmatic manner to drive continual improvement in the SHE risks profile of the company |

**Table A1.** *Cont.*

| She Capability Attributes | Integrated Safety, Health and Environmental Management Capability Maturity Model (iSHEM-CMM) | | | | |
|---|---|---|---|---|---|
| | Capability Levels | | | | |
| | Level 1 | Level 2 | Level 3 | Level 4 | Level 5 |
| | | | | • Processes and plans for SHE risks management are modelled on best practice risks assessment standards, e.g., ISO 31000 | • SHE risks management processes, procedures and control measures are monitored, reviewed and improved on a regular basis to address changing circumstances and to ensure continuing success |
| Management of Outsourced Personnel | • No structured procedure is used in appointing competent outsourced employees, subcontractors and suppliers with regards to the management of SHE <br> • No structured monitoring and assessment of the performance of outsourced employees, subcontractors and suppliers | • Informal procedure in place but rarely used in appointing competent outsourced SHE employees, subcontractors and suppliers <br> • Rare monitoring and assessment of the performance of outsourced employees, subcontractors and suppliers in respect of SHE management <br> • Procedures are poorly documented and maintained | • Formal procedures in place and used occasionally and reactively appointing competent outsource employees, subcontractors and suppliers. <br> • Occasional and reactive assessment of the performance of outsourced employees, subcontractors and suppliers in respect of SHE management <br> • Procedures are adequately documented and maintained | • Regular and proactive procedures are in place for appointing competent outsource employees, subcontractors in a consistent manner <br> • Regular and proactive assessment of the performance of outsourced employees, subcontractors and suppliers in respect of SHE management <br> • All competency definitions are explicitly defined and include industry recognised best practice <br> • Procedures are accurately documented and maintained | • There is a well-structured procedure for appointing, monitoring and assessing the performance of outsourced personnel, subcontractors and suppliers <br> • The well-structured and clear competence management system is integrated within the company's performance of SHE management. <br> • Competence and performance assessment procedures are reviewed regularly to ensure their current suitability and continuous improvement. |
| She Operational Control | • No procedures for identification of SHE operations and activities that need to be controlled to ensure risk associated with them are minimised or eliminated <br> • SHE risks control measures are not in place | • Informal procedures are in place for identification of SHE operations and activities that need to be controlled to ensure risk associated with them are minimised or eliminated <br> • SHE controls measures, are unclear and poorly documented | • Formal procedures are in place for the identification of SHE operations and activities that need to be controlled <br> • Control measures for identified SHE risks are more detailed and clearly stated <br> • Operation control procedures and measures are adequately documented | • Formal and comprehensive procedures are in place for the identification of SHE operations and activities that need to be controlled <br> • Control measures for identified SHE risks are comprehensive and well defined <br> • Identified SHE operations that needs to be controlled and their associated control measures are appropriately documented and well communicated to relevant employees (e.g., suppliers, contractors and other interested parties) | • Well-structured procedures are in place for identification of SHE operations and activities that need to be controlled to ensure compliance, and to achieve objectives <br> • Documented SHE control procedures and measures are continually reviewed and improved |
| She Emergency Preparedness and Response | • No emergency preparedness and response (EPAR) procedures <br> • No measures for identification of possible emergencies and SHE accidents, and how to respond if they arise | • Undefined and inappropriate EPAR procedures and measures for identification of possible emergencies and SHE accidents, and how to respond if they arise <br> • EPAR procedures and measures are poorly documented and are not accessible <br> • Employees are rarely trained in emergency responses | • Defined procedures and measures are available for identification of possible emergencies and SHE accidents, and how to respond if they arise <br> • EPAR procedures and measures are adequately documented but are not easily accessible <br> • Employees are trained in formal emergency responses | • Well-defined and sufficient EPAR procedures and measures for identification of possible emergencies with a focus on specific emergency situations <br> • EPAR procedures and measures are appropriately and accurately documented <br> • EPAR procedures and measures are communicated and accessible to all employees involved <br> • Employees are adequately trained in emergency responses | • Appropriate and comprehensive EPAR plans, procedures and measures are in place to effectively respond to emergency situations <br> • EPAR plans and procedures are fully integrated with other control measures and benchmarked consistently against best practices <br> • EPAR plans are periodically tested for the adequacy of the plan and the results reviewed to improve its effectiveness for continuous improvement |

Table A1. *Cont.*

| | **Integrated Safety, Health and Environmental Management Capability Maturity Model (iSHEM-CMM)** | | | | |
|---|---|---|---|---|---|
| **She Capability Attributes** | **Capability Levels** | | | | |
| | **Level 1** | **Level 2** | **Level 3** | **Level 4** | **Level 5** |
| She Performance Monitoring and Measurement | • No performance measuring and monitoring system in place<br>• SHE procedures for performance monitoring and measurement (MaM) are not well developed<br>• SHE performance indicators and measures are not established<br>• SHE system performance is poor | • There are vague procedures for MaM of SHE performance<br>• Some SHE performance indicators and measures are in place but are not well defined<br>• Performance MaM are rarely undertaken<br>• Some employees are aware of the SHE performance measures in their areas of responsibilities<br>• SHE system performance is fair | • SHE performance MaM procedures and performance indicators and other measures are in place and defined<br>• Performance MaM are undertaken occasionally.<br>• Monitoring is reactive<br>• More employees are aware of the SHE performance measures in the areas of responsibilities<br>• SHE system performance is mostly good | • Well-defined and appropriate performance procedures, key SHE performance indicators and other measures are in place to monitor SHE performance<br>• Performance monitoring and measurement are undertaken regularly with the purpose of improving the SHE system<br>• Performance MaM procedures and measures are compliance led and used to track SHE performance<br>• MaM procedures and measures are adequately documented and communicated to all employees<br>• Employees at all levels are aware of the critical SHE performance measures in their areas of responsibility.<br>• SHE system performance is very good and is constantly repeated | • Well-designed and defined proactive procedures and measures for monitoring, measuring and recording of SHE performance on a regular basis is in place and is institutionalised within the company, focusing on operational excellence and continuous improvement<br>• Results of SHE performance MaM are documented and effectively communicated throughout the company, to facilitate subsequent corrective and preventive actions analysis<br>• SHE performance MaM procedures and measures are continuously used to improve the SHE management system. Best practice is shared across the entire company.<br>• SHE performance of the MaM system is periodically reviewed and improved to make sure they remain relevant to the company's risk profile<br>• SHE system performance is exemplary and comparable to the best in the industry |
| She Incidents Investigations | • No structured processes and procedures for SHE incidents investigations<br>• No organised evidence of SHE investigations | • Vague processes and procedures for SHE incident investigations are in place<br>• The range of incidents investigated is limited to immediate causes of accidents and environmental aspects<br>• Limited employees' involvement<br>• SHE investigations processes and procedures are not documented | • Formal processes and procedures for SHE incident investigations are in place<br>• Investigations tend to focus on the immediate and root causes of SHE incidents, near misses and environmental aspects and their impacts<br>• Incident investigations tend to be reactive<br>• More employees' involvement in SHE investigations.<br>• SHE incident investigations processes and procedures are somewhat documented | • Formal comprehensive and standard processes and procedures for SHE incident investigations<br>• Incident investigations are proactive and probe more deeply to identify direct and indirect causes of SHE incidents and environmental aspects that result in significant SHE risks<br>• Greater employees' involvement in SHE incidents investigations<br>• SHE incidents investigations procedures are communicated to relevant committees for appropriate recommendations and actions<br>• SHE investigation processes and procedures are well documented and corrective actions well communicated to best utilise any lessons to be learned. | • There are documented structured processes and procedures in place for consistently high quality SHE incident investigations<br>• SHE incident investigation procedures are linked to SHE hazard identification and risk a mitigation process and are institutionalised within the company<br>• Outcomes of SHE incidents investigations are seen as opportunities for improvement, and are documented, monitored and shared with industry. SHE incident trends are used to identify and help manage SHE risks<br>• Lessons learned from incidents investigations are shared and implemented across the company.<br>• Corrective and preventive actions are reviewed regularly and updated to ensure actions taken are effective.<br>• SHE incidents investigations procedures are routinely reviewed and updated to drive continuous improvement |

Table A1. *Cont.*

| | Integrated Safety, Health and Environmental Management Capability Maturity Model (iSHEM-CMM) | | | | |
|---|---|---|---|---|---|
| **She Capability Attributes** | **Capability Levels** | | | | |
| | **Level 1** | **Level 2** | **Level 3** | **Level 4** | **Level 5** |
| She System Audits | • No auditing of SHE system<br>• No clear SHE audits processes and procedures | • Company rarely undertake planned SHE system audits. Ad hoc audit with no follow up.<br>• SHE audits processes and procedures are not defined and may not be documented.<br>• Procedures for assessing SHE compliance is limited<br>• Legal and regulatory obligations noncompliance | • Company occasionally undertakes planned SHE system audits<br>• SHE audit processes and procedures are somewhat defined and poorly documented<br>• Most aspects of the SHE system are audited with some follow-up<br>• Minimal legal and regulatory compliance<br>• SHE audits processes and procedures are focused on achieving compliance with legal and regulatory obligations | • Company regularly undertakes planned SHE audits<br>• SHE audits processes and procedures are well defined and designed, and modelled on best practice of audits<br>• All aspects of SHE system audited with some follow-up<br>• Total legal and regulatory obligations compliance Written recommendations, (e.g., non-compliances) are well documented and communicated to form the basis of SHE improvement and innovation.<br>• SHE audits processes and procedures are modelled on best practice standards for auditing management system, e.g., ISO 19011:2018 guidelines for auditing management systems, OHSAS 18001:2007 | • There is a company-wide standardised audit system in place and institutionalised within the company, with best practice shared internally with other functions of the company<br>• SHE audits are undertaken regularly by competent employees to demonstrate compliance with required standards, legal and regulatory obligations.<br>• SHE audits processes and procedures are planned and prioritised, and covers all aspects of the SHE system.<br>• SHE audits process and procedures are reviewed periodically to ensure they are current and consistent with leading internal audit practice and standard requirements in order to ensure continuous improvement in audit processes |
| Roles and Responsibilities for She | • No clear SHE roles, and responsibilities (i.e., there are no roles, tasks and objectives given to people and teams to meet the organisation's SHE objectives) | • SHE roles and responsibilities are unclear with some specific responsibilities and authorities somewhat defined and developed.<br>• SHE roles and responsibilities are not recorded in job descriptions | • SHE roles and responsibilities are mostly defined and assigned to employees<br>• SHE roles and responsibilities are inconsistently recorded in job descriptions | • SHE roles and responsibilities are well defined, sufficiently comprehensive and well communicated to designated employees at all levels<br>• All SHE roles and responsibilities are consistently recorded in key documentation (e.g., job descriptions) and appropriate communication media | • Clearly defined SHE roles, responsibilities and authorities at all levels of the company<br>• SHE roles and responsibilities are unambiguous, clearly understood and accurately documented<br>• SHE roles, responsibilities and authorities are continuously reviewed, realigned to effort and tracked to ensure proper distribution and continuous improvement |
| She Training | • No provision of SHE-related training for employees<br>• No formal training needs analysis undertaken | • Provision of SHE-related training for employees is very low and unplanned. Provision of SHE training is rarely informed by a formal training needs analysis<br>• Training needs are not well defined and documented | • Provision of SHE-related training is reactive<br>• Provision of SHE training is occasionally informed by a formal training needs analysis<br>• Identified training needs are somewhat defined and based on the wider competency and performance objectives<br>• Training needs adequately documented | • Regular provision of adequate SHE-related training for employees, informed by a formal and objective training needs analysis undertaken on a regular basis<br>• Training is typically based on employees SHE roles and respective competency objectives<br>• Training needs are well defined and accurately documented (e.g., in the employees' personal files) | • Appropriate and timely SHE training is in place and is integral to company's human resource strategy to improve SHE performance<br>• SHE training strategies are incorporated into the company's overall, SHE management strategies and policies<br>• SHE-related training programmes or plans are reviewed for their effectiveness and are periodically reviewed to ensure their current suitability.<br>• SHE-related training programmes and training are continuously assessed and updated to reflect organisational, regulatory changes and any other changes in technology and techniques, to allow continuous learning and improvement |

**Table A1.** *Cont.*

| She Capability Attributes | Integrated Safety, Health and Environmental Management Capability Maturity Model (iSHEM-CMM) | | | | |
| --- | --- | --- | --- | --- | --- |
| | Capability Levels | | | | |
| | Level 1 | Level 2 | Level 3 | Level 4 | Level 5 |
| | | | | • Training is usually proactive, tracked and evaluated to be improved upon | • The various training methods are incorporated into the knowledge and communication channels of the company<br>• Training needs analysis procedures are regularly reviewed |
| Employee Involvement In She | • No consultation of employees on SHE-related issues<br>• Employees are not involved and have no interested in participating in SHE related issues | • Limited consultation on SHE-related issues, but not carried out in a systematic way. Minority of the employees are involved and interested in participating in SHE-related issues | • More consultation on SHE issues is carried out in a systematic way<br>• Majority of the employees are involved and interested in participating in SHE-related issues | • All employees are regularly consulted on SHE-related issues and are carried out in a range of ways (e.g., surveys, workshops, site meetings and committees)<br>• Overwhelming majority of the employees are involved and interested in participating in SHE-related issues<br>• Employee involvement and consultation arrangements are documented and interested parties are informed | • All employees are fully consulted and actively engaged in SHE related issues at all company's levels.<br>• All employees are interested in participating SHE related issues<br>• Company use employee involvement to gather ideas for improvement on SHE issues<br>• Company makes full use of employees' potential to develop shared values and a culture of trust, openness and empowerment |
| She Competence | • Company's employees do not have the skills, knowledge and the experience necessary for SHE management | • An overwhelming majority of company's employees have basic SHE knowledge and skills, with no employees having advanced or expert skills and knowledge<br>• Company's employees have limited experience in SHE management tasks | • A majority of company's SHE employees have intermediate SHE skills and knowledge with very few having advanced and/or expert skills and knowledge<br>• Company's employees have some experience in SHE management tasks | • A majority of company's employees have sufficient and advanced SHE skills, and knowledge with very few having basic or no SHE skills and knowledge<br>• Company's employees have appropriate experience in SHE management tasks | • An overwhelming majority of company's employees have expert SHE skills and knowledge with very few or none having basic or no SHE skills and knowledge<br>• Company's employees have vast and experience in SHE management tasks<br>• Company's employees feel competent and capable to perform their SHE tasks. |
| Physical She Resources | • No physical resources available to enable SHE employees to perform SHE-related tasks. | • Company is ill-equipped with physical resources for employees to perform SHE-related tasks. Physical SHE resources are limited<br>• Resource provision is not or rarely informed by any strategic resource plan | • Company is equipped with adequate physical SHE resources to enable employees to perform SHE-related tasks.<br>• Resource provision is usually reactive and occasionally informed by a strategic resource plan | • Company is well equipped with sufficient physical resources for employees to perform SHE-related tasks.<br>• A strategic resource plan is available to inform timely provision of physical resources to enable employees to perform SHE-related tasks | • Company is fully equipped with sufficient resources in quality and quantity for employees to perform SHE-related tasks<br>• Company's SHE physical resources are considered to be integral to SHE performance and competitiveness<br>• Physical resources are continuously tested, upgraded and deployed<br>• Resource plans for provision of physical resources are documented and integrated into company's processes and systems to improve effectiveness and efficiency<br>• Resource plans are regularly reviewed to ensure the provision of adequate and current resources to meet planned and agreed targets and objectives |

**Table A1.** *Cont.*

| Integrated Safety, Health and Environmental Management Capability Maturity Model (iSHEM-CMM) | | | | | |
|---|---|---|---|---|---|
| Financial Resources For She | • No financial resources for SHE implementation.<br>• Unstable or uncertain funding | • Limited financial resources for SHE implementation and rarely informed by a strategic resource plan<br>• No established sources of funding | • Company has adequate financial resources for SHE implementation<br>• Provision of financial resources is occasionally informed by strategic resource plan<br>• Established source of funding | • Company has sufficient and well organised funding lines for SHE implementation.<br>• A strategic resource plan is available to inform timely provision of financial resources for effective SHE management<br>• Stable sources of funding | • Dedicated and adequate financial resources in place for effective SHE implementation and considered to be an integral part of the company's finance plan<br>• Highly stable funding. Resource plans are regularly reviewed to ensure the provision of adequate and current resources to meet planned and agreed targets and objectives |
| She Communications | • No formal communication of any SHE-related issues to employees<br>• No formal communication channels for effective flow of SHE information internally and externally in the company | • Limited communication of SHE information to employees.<br>• Communication is ad hoc and restricted to those involved in specific incidents.<br>• Company's employees are unaware of important SHE information<br>• Some informal and formal communication channels are established for information flow internally to all employees. | • Some communication of SHE information to employees on a need-to-know basis<br>• There is a communication strategy for SHE information flow internally and externally occasionally to all employees.<br>• Employees are aware of pertinent SHE information.<br>• Specific informal and formal communication channels are in place for communicating SHE issues to employees | • Adequate SHE information is routinely and regularly communicated to all employees. Employees are aware of critical SHE information.<br>• There are established, good and appropriate informal and formal communication channels for communicating critical SHE information and resultant actions<br>• All levels of employees are involved, and there are robust mechanisms for them to feedback | • There is an open, proactive and effective SHE communication between the company and its employees and stakeholders.<br>• SHE communication is a strong, and consistent two-way process. Good practice is communicated both externally and internally<br>• The company communicates to its employees on all the SHE-related issues and aspects of the company.<br>• Established communication channels and methods are fully adopted throughout the supply chain in the company and are consistently used for efficient coordination of SHE activities.<br>• All pertinent SHE information and resultant actions are well communicated to all employees across the company.<br>• Communication methods for SHE information flow internally and externally are continuously monitored and regularly reviewed against identified best practices in other sectors for potential continuous improvement. |
| She Documentation and Control | • No organised documentations (e.g., SHE policy, SHE manual, emergency plans and work instructions etc.) and records that describe company's SHE system elements and their interrelationships | • Documentations of some elements of a company's SHE system and other related SHE records are available to employees<br>• SHE documentations and records are not organised and are not easily traceable and accessible | • Documentations and records of more elements of a company's SHE system and other related SHE records are available to employees<br>• SHE documentations and records are compiled and organised in a format that is somewhat traceable and accessible | • Documentations and records of all elements of the company's SHE system and other related SHE records are available to all employees<br>• All SHE documentations are compiled and mostly organised in an appropriate format, traceable and accessible. | • SHE documentations including other related SHE records are compiled and well organised in a clear, concise and functional format, traceable and readily accessible to all.<br>• SHE documentations and records are integrated with other organisational documentations (such as human resource plans) for continuous improvement of company's functions.<br>• SHE reports and SHE documentations are systematically maintained regularly reviewed and updated with appropriate version control in place, based on system improvements, to drive efficiency and effectiveness of the management system. |

**Table A1.** *Cont.*

| | Integrated Safety, Health and Environmental Management Capability Maturity Model (iSHEM-CMM) | | | | |
|---|---|---|---|---|---|
| **She Capability Attributes** | **Capability Levels** | | | | |
| | **Level 1** | **Level 2** | **Level 3** | **Level 4** | **Level 5** |
| Lessons learned and knowledge Management | • Company has no structured system for capturing lessons in order to facilitate future improvement of the SHE management system<br>• No promotion of knowledge sharing and lessons learned across the company<br>• No records of lessons learned. There is great reliance on individual memory | • Company's processes and procedures for capturing and disseminating lessons learned are characterised by poor or unstructured records keeping and inconsistent data<br>• Limited promotion of knowledge sharing and lessons learned across the company<br>• Reliance on manual record keeping of lessons<br>• Lessons learned are rarely used for SHE management system continuous improvement and innovation | • Company's processes and procedures for capturing and disseminating lessons learned are characterised by well-structured record keeping and good information<br>• Knowledge sharing and lessons learned is promoted across the company<br>• Little reliance on manual record keeping and greater usage of digital technologies for record keeping<br>• Records of lessons learned are sometimes relied on for SHE management system continuous improvement and innovation | • Company's processes and procedures for capturing and disseminating lessons learned are characterised by routinely well-structured record keeping and consistent high-quality information<br>• Knowledge sharing and lesson learned is promoted systematically across the company<br>• Reliance on advanced digital technologies for capturing and disseminating lessons<br>• Records of lessons are consistently relied upon for SHE decision making, continuous improvement and innovation<br>• Processes and procedures for capturing and disseminating lessons learned are modelled on best practice knowledge management standards e.g., ISO 30401-2018, ISO 9001: 2015. | • There is well structured system for capturing and disseminating lessons learned and knowledge gained across the whole company. Heavy reliance on technological innovations for capturing and disseminating lessons<br>• The processes are institutionalised within the company and are considered a key measure of operational excellence.<br>• Knowledge and lessons learned are continuously shared and consistently relied upon across the company to continuously improve SHE<br>• Processes and procedures for capturing and disseminating lessons learned are routinely reviewed and updated to drive continuous improvement and innovation. |

**Table A2.** Evaluation Questionnaire.

| Assessment Criteria | Level of Agreement | | | | |
|---|---|---|---|---|---|
| | **Strongly Agree** 5 | **Agree** 4 | **Neither Agree nor Disagree** 3 | **Disagree** 2 | **Strongly Disagree** 1 |
| Attributes used in the SHEM-CMM Worksheet | | | | | |
| Attributes are relevant to SHE management capability. | ☐ | ☐ | ☐ | ☐ | ☐ |
| Attributes cover all aspects of SHE management capability. | ☐ | ☐ | ☐ | ☐ | ☐ |
| Attributes are correctly assigned to their respective maturity level. | ☐ | ☐ | ☐ | ☐ | ☐ |
| Attributes are clearly distinct. | ☐ | ☐ | ☐ | ☐ | ☐ |
| *Capability Maturity Levels* | | | | | |
| The maturity levels sufficiently represent maturation in the attributes. | ☐ | ☐ | ☐ | ☐ | ☐ |
| There is no overlap detected between descriptions of maturity levels. | ☐ | ☐ | ☐ | ☐ | ☐ |
| *Ease of Understanding* | | | | | |
| The maturity levels are understandable | ☐ | ☐ | ☐ | ☐ | ☐ |
| The documentations (i.e., assessment instructions) are easy to understand | ☐ | ☐ | ☐ | ☐ | ☐ |
| The results are understandable | ☐ | ☐ | ☐ | ☐ | ☐ |
| *Ease of Use* | | | | | |
| The scoring scheme (i.e., drop-down options for maturity levels (1–5)) is easy to use | ☐ | ☐ | ☐ | ☐ | ☐ |
| The SHEM-CMM is easy to use | ☐ | ☐ | ☐ | ☐ | ☐ |
| *Usefulness and Practicality* | | | | | |
| SHEM-CMM is useful for assessing SHE management capability | ☐ | ☐ | ☐ | ☐ | ☐ |
| SHEM-CMM is practical for use in industry | ☐ | ☐ | ☐ | ☐ | ☐ |

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
