# Peer review of "An Integrated Safety, Health and Environmental Management Capability Maturity Model for Construction Organisations: A Case Study in Ghana"

_buildings, doi:10.3390/buildings11120645_

Round 1
Reviewer 1 Report
The present study concluded that the 19-maturity model is a useful assessment framework or tool for industry stakeholders, particularly 20 construction firms, to evaluate the status of their current SHE management capability, identify 21 strengths and improvement areas, and accordingly priorities strategies/actions for improving their 22 SHE management.
The main contribution, novelty and comparison is missing in the paper. (The authors mentioned that This study thus examines integrated SHE management capability)
Please add that at the 2nd last paragraph of your introduction)
In the literature regarding - Safety, health and environmental management capability in construction. How did you conducted keywords search and mention relevant sources (Google Scholars, WoS, Scopus)
Conclusion section can be improved by highlighting key findings, limitation of the research and recommendation for future studies.
Methods used in the study needs to be compared well with the efficiency of one method over the other being compared statistically, I can see that missing clearly.
Explain why the current method was selected for the study, its importance and compare with traditional methods.
Also add risk factors in your study – risk associated with this domain – risk matrix.
Reviewer 2 Report
Paper demonstrates an integrated Safety, Health and Environmental Management Capability Maturity Model for Construction Organizations. The novelty of this paper is significant enough to publish on “Buildings”.
Due to some issues that need to improve before publication, thus my decision is an acceptance with minor revisions. Here are my comments for improving the manuscript:
- Abstract: The authors stated that “However, there is limited empirical insight regarding integrated SHE management capability of construction companies”. However, many international construction firms have applied this maturity model successfully. Please explain.
- Introduction:
- The introduction provides background information and set the context. However, introduction does not bring the problem to the forefront of what we did not yet know and WHY the authors needed to study it. Thus, please introduce the specific topic of your research and emphasize why it is important.
- Please state research questions and research objectives. Authors can mention past attempts to solve the research problem or to answer the research question, then conclude the introduction by mentioning the specific objectives of your research.
- Materials and Methods:
- In comparison with step 5, 6, and 7, the explanation of step 1, 2, 3, 8, 9, and 10 are quite poor. Please describe more detailed.
- Results and discussion:
- The model is validated by SHE experts from 59 construction firms operating in Ghana. Due to this research limitations, thus please consider to change the title to “An Integrated Safety, Health, and Environmental Management Capability Maturity Model for Construction Organizations: Case study in Ghana”.
- Please kindly provide a detailed questionnaire to survey
- Sample for validation is small, thus result is not reliable enough to validate the proposed maturity model. Please explain.
- Conclusion:
- Beside emphasizing the research contributions, conclusion would be improved by pointing out research limitations that motivate to study future works
- Due to a lack of 5.2 section, please remove 5.1 numbering
- References:
- Some references are too old. Please replace the new ISI ones to improve quality of this reference section.
